# Sovereign Wealth Funds as Sustainability Instruments? Disclosure of Sustainability Criteria in Worldwide Comparison

Stefan Wurster *[ID] and Steffen Johannes Schlosser

TUM School of Governance, Technical University Munich, 80333 Munich, Germany
* Correspondence: stefan.wurster@hfp.tum.de

**Abstract:** Sovereign wealth funds (SWFs) are state-owned investment vehicles intended to pursue national objectives. Their nature as long-term investors combined with their political mandate could make SWFs an instrument suited to promote sustainability. As an essential precondition, it is important for SWFs to commit to sustainability criteria as part of an overarching strategy. In the article, we present the sustainability disclosure index (SDI), an original new dataset for a selection of over 50 SWFs to investigate whether SWFs disclose sustainability criteria covering environmental, social, economic, and governance aspects into their mandate. In addition to an empirical measurement of the disclosure rate, we conduct multiple regressions to analyze what factors help to explain the variance between SWFs. We see that a majority of SWFs disclose at least some of the sustainability criteria. However, until today, only a small minority address a broad selection as a possible basis for a comprehensive sustainability strategy. While a high-state capacity and a young population in a country as well as a commitment to the international Santiago Principles are positively associated with a higher disclosure rate, we find no evidence for strong effects of the economic development level, the resource abundance, and the degree of democratization of a country or of the specific size and structure of a fund. Identifying favorable conditions for a higher commitment of SWFs could help to initiate pathways to become functional sustainability instruments.

**Keywords:** sovereign wealth funds; long-term investing; sustainability; sustainability disclosure index; ESG principles; sound governance; state capacity; demography; Santiago Principles

## 1. Introduction

International markets and large-scale investors have a critical influence on the achievement of sustainability. The concept of sustainability is understood, in this research, as the capability to globally meet "the needs and aspirations of the present generation without compromising the ability of future generations to meet their need" [1] (p. 292) while avoiding the transgression of "planetary boundaries" [2]. Large-scale investors can invest in an unsustainable way for the sake of short-term financial profits and undermine sustainability as understood in the present work. On the other hand, large-scale investors could also become a catalyst for the development toward sustainability since they have the financial capacities to actively promote specified sustainability goals. An active promotion of sustainability requires large-scale investors to explicitly regard sustainability considerations in their investment decisions and to actively search for suitable investment opportunities. The active search for sustainable investment opportunities is expected to be formulated into commitments in the publicly accessible official documents of these investors.

The present research is focused on the regard of sovereign wealth funds (SWFs) [3,4] —large, state-owned investment entities as a special instance of large-scale investors—for sustainability. State-owned sovereign wealth funds as well as public pension funds had over USD 27 trillion in assets under management in 2020, which makes these funds the third largest group of asset owners, globally [5] (p. 1). SWFs are liable to the governments of their countries of origin and are intended to achieve national objectives

rather than being limited to following the capital interests of shareholders. SWFs can be expected to be influenced by the respective political and economic systems of their countries of origin, while long-term orientation is a strong precondition for its institutional self-preservation.

Against this background, the question about the potential of SWFs to promote sustainability has been raised for several years (see inter alia [6,7]); however, it gained new momentum since the UN introduced the Sustainable Development Goals in 2015, promoting the concept of responsible allocation and investment [8]. The concept of sustainability includes multiple sustainability criteria, which can be understood as measurable indicators for its different aspects. A set of sustainability criteria, which is often referred to, are the Sustainable Development Goals (SDGs) defined by the United Nations (UN). Liang and Renneboog stated in 2020 that it is reasonable to expect SWFs to pursue sustainability due to their nature as long-term investors in charge of preserving wealth for future generations [9]. Further, focusing exclusively on the sustainability goal of climate protection, there are approaches for strategies that aim to enable "long-term passive investors-a huge institutional investor clientele that includes [...] sovereign wealth funds-to significantly hedge climate risk while essentially sacrificing no financial returns" [10] (p. 14). This growing branch of literature [6,7,9,10] implies that it is important to investigate the contemporary relation of SWFs and sustainability in a worldwide empirical comparison. However, to date, a big gap in the existing research literature exists regarding the empirical measurement of sustainability commitments by SWFs as well as the quantitative analysis of factors influencing their behavior.

This research is intended to overcome this gap and explore which specified sustainability criteria, covering environmental, social, economic, and governance aspects, are regarded by SWFs in the year 2020. For this purpose, we present an original, new dataset, the Sustainability Disclosure Index (SDI), covering the disclosure of 19 sustainability criteria for a selection of 68 SWFs that manage more than 90 percent of all assets under management (AUM) by SWFs (see a detailed list in supplementary A4). Using it, we can evaluate whether singular sustainability criteria are disclosed sporadically or whether the disclosed sustainability criteria can be considered as a part of an overarching sustainability framework covering ecological as well as economic, social, and governance aspects. Since it is not clear whether SWFs react due to internal functional logic (long-term self-preservation) or due to external incentives and pressure, the impact of the political and socioeconomic aspects of the countries of origin and international factors as well as fund-specific characteristics on the overall disclosure of sustainability criteria is investigated. Therefore, two research questions are pursued.

(1)　Which sustainability criteria are disclosed by SWFs? What roles do environmental, social, economic, and governance aspects play, and how far are overarching sustainability frameworks observable?

(2)　In what way do the political and socioeconomic aspects of their respective countries of origin, international factors as well as fund-specific characteristics influence the disclosure of sustainability criteria by SWFs?

The present research paper consists of six parts. After the introduction, we present the theoretical background in the following Section two, focusing on the specifics of SWFs and the theoretical expectations regarding their (un)sustainable investment strategies. We also formulate specific hypotheses about the influence of socio-economic, political, international, and fund-specific factors regarding the (non)consideration of sustainable criteria by SWFs. Section three gives an overview regarding the construction of our original SDI dataset (dependent variable), the operationalization of the independent variables, and describes our methodical approach. We present the empirical results regarding the disclosure of sustainability criteria into the mandate of SWFs (descriptive analysis of the dependent variables; in-depth analysis of SWFs following an overarching sustainability strategy) and the multiple linear regression analysis in Section four. From our original sample of funds up to ten cases, missing data points for the independent variables had to be omitted from

the regression analysis. Data missingness for the AUM and the origin of funding is relevant for the Common School Fund of Oregon, the Endowment Fund Investment of Idaho, the Fourth Swedish National Pension Fund, the National Investment Corporation of Abu Dhabi, the Sharjah Asset Management, the Turkey Wealth Fund, and the West Virginia Future Fund. These funds had to be omitted from the fund-specific models and full models in our regressions. Data missingness for the GDP/capita was relevant for Brunei, Palestine and Timor-Leste, so the Brunei Investment Agency, the Palestine Investment Fund and the Turkey Wealth Fund had to be omitted from the socioeconomic and political models and the full models in our regression. To validate our regression results for the full model, including all independent variables, we conduct an additional analysis with reduced samples of independent variables (fund-specific model, socioeconomic and political model) and do some robustness checks with different independent variables. See for a flowchart of our methodological approach Figure 1.



**Figure 1.** Flowchart of the methodological approach.

After the discussion of the results in chapter five, we answer our research questions in the concluding chapter six, also discussing the limitations of our study and the need for further research. The innovative value of our contribution lies in the fact that, based on the SDI, we can make sustainability commitments of SWFs measurable in a differentiated way and we offer, to our knowledge, the first study that includes a quantitative regression analysis of possible influencing factors.

## 2. Theoretical Background

Sovereign wealth funds (SWFs) can generally be defined as "pools of assets owned and managed directly or indirectly by governments to achieve national objectives" [4] (p. 4). A more specific definition by Fotak et al. specifies SWFs as "an investment fund rather than an operating company, that is wholly owned by a sovereign government, but organized separately from the central bank or finance ministry to protect it from excessive political influence, that makes international and domestic investments in a variety of risky assets, that is charged with seeking a commercial return" [3] (p. 2). According to Blundell-Wignall et al. [4], conceivable objectives of SWFs include the diversification of assets, the optimization of the return on reserves, the provision of pensions in the future (pension funds), providing for future generations after the depletion of natural resources (savings funds), stabilizing prices (stabilization or "rainy-day funds" as a buffer mechanism to cover fiscal deficits in times of crisis), facilitating industrialization, and attempting to achieve strategic and political objectives (strategic development funds, economic mission contributing to the development of the domestic economy; [4] (p. 4)). SWFs mainly originate from revenue from the sale of natural resources, accumulated foreign currency reserves, export surpluses or pension contributions [3] (p. 9). Importantly, SWFs do not have explicit short-term liabilities and are more focused on long-term investments. From an economist's point of view, SWFs are long-term, passive investors [10]. This feature could make SWFs suited to sustainable investing practices since unsustainable investing practices could undermine the foundations which SWFs are based on in the long run (institutional self-preservation).

"Sustainable investing" can be understood as investments suited to promoting sustainability, keeping in mind that also, "sustainable investments" contribute to an overall expansive economy. It is crucial that sustainable investments grow faster than unsustainable investments and can replace them within a foreseeable timeframe. Having said this,

one should note that sustainability promotion is a very complex task since it means to expand responsibility from "here and now" to "there and later", adapting and balancing current and future interests [11]. As a multi-dimensional, cross-sectional mission, it requires taking into account different policy objectives and criteria in parallel. We understand sustainability as having three pillars, which are the environment, the society and the economy [12,13]. Further, good governance practices are crucial for sustainability [14] and can be understood of as its fourth pillar. However, it is not enough just to pursue one single sustainability pillar or goal. Instead, all goals must be pursued in a balanced manner (overcoming conflicting targets) and should be integrated into a holistic, overarching sustainability framework. In addition to the increased complexity of long-term decision making which requires actions under the conditions of fundamental uncertainty [15] (the further we look into the future, the more unclear are the effects of current decisions since unintended consequences and reflexive social behavior makes foreseeing possible effects a difficult task [11]), taking sustainability seriously is a normative task since it claims to create intergenerational and intragenerational justice [16]. "Intergenerational justice requires taking the interests of future generations into account and to overcome discounting incentives of current generations, while intragenerational justice requires to consider the needs of distant people and to overcome international cross border problems like climate change, invasive species, etc." [11].

Even though there are incentives for SWFs to take certain sustainability aspects and criteria seriously, especially if this results in new market access and income opportunities (for example, regarding the promotion of renewable energy), the complex normative and practical challenges described above make it very difficult for SWFs to commit themselves to a comprehensive, overarching sustainability strategy. We expect that their engagement (if any) will be limited to commitments that do not harm their immediate investment opportunities and capacity to act but rather helps them to become more competitive and, at the same time, increase their confidence and reputation among other actors. In particular, commitment with regard to responsible, transparent, risk-sensitive, and future-oriented governance criteria might be an asset in this regard [5].

Concerning our first research questions, we therefore formulate two hypotheses:

**Hypothesis 1.** *SWFs do not follow an overarching sustainability strategy, meaning a disclosure of a broad majority of sustainability criteria covering all four sustainability pillars into their mandate.*

**Hypothesis 2.** *If SWFs disclose sustainability criteria, they prefer governance criteria.*

What factors might facilitate (or hinder) a SWF committing itself to pursue sustainability goals and integrating them into an overarching sustainability strategy? Besides the internal structures and characteristic of a fund, which might pull toward long-term, sustainable investment as part of a rational self-preservation strategy, external push factors might "force" a fund to move in this direction. In the following, we will introduce fund-specific, socio-economic and political as well as international factors, which might be relevant.

*2.1. Structure of the Fund (Pension Fund Mandate or Other SWFs)*

If we first look at the structure of the fund, the distinction between pension funds and other SWFs appears to be decisive since they have different timeframes for their investments. Whereas pension funds have to generate high immediate revenue to finance pensions (liabilities of pensions must be serviced), other SWFs are much more flexible with regard to their investing. As rarities in the investing world in being almost totally unconstrained in their investment policies, they face little or no pressure to boost short-term returns, which facilitates a sustainable investment orientation [5] (p. 3). The quite strict transparency requirements, in terms of their investment policies and allocations to which pension funds are subject, cannot compensate for this advantage, since strict orientation on the present interest of the (upcoming) retirees is a crucial part of their self-preservation strategy. In addition to the inspection rights of the pension funders, many pension funds

are also subject to electoral oversight since a vast majority of them are based in democratic societies [5] (p. 7). While open processes in democracies force many pension funds to be quite transparent, short-term imperatives and desires in democracies have led a great many of them to be underfunded regarding their promised long-term payouts to retirees [17]. Accordingly, we expect that ceteris paribus other SWFs are more suited to sustainable investing as compared to pension funds.

**Hypothesis 3.** *Non-pension funds disclose more sustainability criteria into their mandate than pension funds.*

### 2.2. Size of the Fund

A second important structural characteristic of a fund is its size, measured by its AUM. The variance between the funds included into this study is enormous, ranging from hundreds of billions (USD) to hundreds of millions (USD) [18,19]. We argue that funds of a larger size are expected to have more room for sustainable investing as compared to funds of a smaller size. The revenue generated by the fund is a convolution of its capital stock and its return. Accordingly, if funds with a high capital stock accept lower returns for the purpose of their investments being sustainable, they might still achieve revenue high enough to fulfil their mandate. However, it is possible that this does not work for funds with a smaller capital stock. Furthermore, funds of a larger size are subject to higher public scrutiny and might, therefore, face pressure to invest more sustainably, whereas smaller funds are not surveyed as closely, so unsustainable behavior might go unnoticed.

**Hypothesis 4.** *The bigger a fund (extent of AUM), the more sustainability criteria it discloses into its mandate.*

### 2.3. Source of Funding of the Fund

A third fund-specific factor that should be considered when thinking about its sustainability orientation is the financial basis of a fund and its (original) source of funding. As mentioned above, the capital stock of a fund can originate from different sources. While still a majority of SWFs are based on revenues from the exploitation and sale of natural resources (oil and gas, and some also from others, such as diamonds or copper), a growing number build up on non-commodity sources. In the literature, we find reference that the shift in purchasing power among SWFs from those ultimately funded through oil and gas sales by state-owned, national companies toward those funds that are financed by the accumulation of foreign currency reserves, export surpluses, pension contributions or government deposits out of other sources have augmented their push toward responsible and sustainable investing [5] (p. 6). The underlying argument is that non-commodity-based SWFs might find it easier to follow a more sustainability-oriented investment strategy since this does not necessarily contradict their previous business model. For funds based on the exploitation of natural resources, it is much harder to credibly convey such an investment orientation since it plunges them into a legitimacy dilemma. Such a reorientation calls into question their previous, at least ecologically, not very sustainable business model and in doing so, could even endanger it. Such a strategic change would entail considerable risks and challenges for them.

**Hypothesis 5.** *Funds with a source of funding other than natural resources disclose more sustainability criteria into their mandate than funds with a source of funding coming out of natural resources.*

### 2.4. Economic Status of Country of Origin

Besides fund-specific factors, the socio-economic environment provided by the country of origin of a fund with its specific incentive and restriction structures should have an impact on the investing orientation of an SWF. One of the key factors that we consider here is the economic development level of a country. We expect that funds located in economically

developed countries have stronger incentives to disclose long-term criteria into their mandate for two reasons. Contrary to the situation in developing countries, frequently putting high presser onto "their" funds to support a national short-term economic growth strategy, the situation in a richer country is different. Here, funds might have the freedom to be invested in a more sustainable way due to lower pressure to generate immediate and high short-term revenue. Therefore, the inclusion of investment criteria addressing intergenerational and intragenerational justice might be easier. While pressure to achieve immediate economic gains relaxes, public pressure to consider sustainability aspects should be higher in richer countries. Following the postmaterialism argument [20], one could expect that if the most pressing material needs of the people are satisfied, postmaterialistic values (such as sustainability orientation) become more important. If we assume that SWFs cannot decouple themselves completely from public expectations, this should have an impact, especially since national governments might be under pressure to commit the funds to corresponding goals.

**Hypothesis 6.** *The higher the economic status of the country of origin of a fund, the more sustainability criteria that it discloses into its mandate.*

### 2.5. Natural Resource Wealth of Country of Origin

In line with the argumentation above, we expect that the natural resource wealth of a country stands in negative relation to the probability of a SWF to consider sustainability criteria. Even so, one could argue that resource-abundant countries might be aware of the foreseeable depletion of their natural resources and might, therefore, be inclined to convert the related financial revenue into a continuous and more stable stream of financial revenue, which comes from sustainable sources; however, strong resistance has to be overcome. Domestic natural resource industries might defend their business model and influence the government to refrain from making SWF a vehicle to overcome concentration on natural resources.

**Hypothesis 7.** *The higher the natural resource wealth of the country of origin of a fund, the fewer sustainability criteria that it discloses into its mandate.*

### 2.6. Demography of Country of Origin

In addition to economic factors, the demographic situation of a funds country of origin could also play a role for the sustainability orientation of a SWF. In particular, societies with a high share of young people are expected to be very aware of sustainability issues because a high percentage of its population will be directly affected by the consequences of today's (un)sustainable behavior in the future. For societies with a higher share of older people, we cannot expect such a strong sustainability orientation since a larger proportion of the population will not directly be affected by the mid- to long-term consequences of their current behavior. Accordingly, we can expect that a high share of young people in a country could put pressure on economic, social, and political decision-makers to take sustainability issues more seriously. Even though very young citizens lack the opportunity to influence political decision-making through their immediate voting decision, they have other opportunities to exert at least indirect influence and generate pressure on the political system. The current protests of the "Friday for future" movement in many countries, which are mainly supported by very young people, show this in an impressive way. Although SWFs are, to a certain degree, shielded from direct political influence, public debates and opinions should have at least an indirect effect on the decision-making processes. While in older societies public support to prioritize investments with high short-term revenues might be higher, in younger societies, public pressure and appropriate initiatives and influence efforts by policy makers to opt for long-term sustainable investment strategies could incentivize corresponding changes in the mandates of the SWFs.

**Hypothesis 8.** *The higher the share of young people in the country of origin of a fund, the more sustainability criteria that it discloses into its mandate.*

### 2.7. Electoral Democracy

Looking specifically at political factors as further possible drivers for the sustainability behavior of SWFs, the political regime from which a SWF originates should come into consideration. Although SWFs are "intentionally separated-either legally or operationally, or both-from other ministries and agencies in order to shield the funds' managers from direct political pressure" [3] (p. 10), the degree of this decoupling is expected to vary throughout the countries since unofficial personal networks might be superimposed over the formally existing relations between the state and SWF. Generally, it is expected that governments are able to exert an influence on the overall agendas of SWFs but might be limited with respect to their implementation. Consistent with this assumption, Bortolotti et al. [21] state that governments might influence the objectives of SWFs (political interference in SWFs, with regard to the incorporation of sustainability criteria's, is well documented [22,23]. The same is true of different party effects on investment strategies of SWFs [5,24]). In the following, we argue that in electoral democracies, characterized by the existence of "contested elections", including public contestation and the right to participate for the citizens of a country (see [25]), SWFs are more likely to include sustainability criteria into their mandate than in autocratic systems. Strong political competition, open public participation, and free speech (including the existence of free media) does not only force governments to justify their actions, giving permanent incentives for policy optimizing (error-correction capabilities) and learning (early warning system) [11,26], but also puts pressure on institutions such as SWFs to make their investment behavior more transparent. This makes it more likely for SWFs to make their mandate publicly available and try to include those other than purely short-term financial goals into it.

In addition, one can assume that democracies generally have a higher capability to react to the demands of their societies than autocracies due to the better discourse practices in democracies and the bigger selectorate [27]. In this vein, democracies are expected to be better equipped to react if their societies demand a stronger regard for sustainability. A possible way to react to the demand for sustainability is to make the SWF investing behavior more aligned with the concept of sustainability. However, it is also in democracies that exists an open question as to what degree the interest of future generations (and people from other countries) is taken into consideration by the national selectorate [26]. This is more likely the more that this goes hand in hand with concrete advantages for the current majority of the selectorate, while in autocracies, this is only likely if the minority supporting the autocratic ruler also benefit from it. The well-known problem in a democracy—its short-lived political time timeframe caused by permanent election atmosphere [26], which could lead to unsustainable economic behavior—does not, however, apply in the specific case of SWFs since their actual investment decisions are, as mentioned above, at least to a certain degree, decoupled from short-term voting cycles.

**Hypothesis 9.** *The more democratic the political regime in the country of origin of a fund, the more sustainability criteria that it discloses into its mandate.*

### 2.8. State Capacity

Besides the political regime type, state capacity meant as "the ability of the state to implement official goals, especially over the actual or potential opposition of powerful social groups" [28,29] (p. 9, p. 741) is crucial to promote sustainability goals. Without administrative skills, it is impossible for a state to implement targeted policies. This applies both for public-based decisions in democracies as well as instructions-based regulations in autocracies [30]. Generally, the strength of the state capacity is not in linear relation to the degree of democratization [31]. Often, a high state capacity is found in advanced democracies as well as in functional autocratic systems. In the context of the Sustainable Development Goals (SDGs), it has been pointed out that a functional state administration

is a necessary requirement to fulfill those [32]. So, a high state capacity is expected to be a necessary precondition regarding the sustainability of the state. States with a dysfunctional state capacity have either no room to consider sustainability as an important topic or to be simply unable to take respective measures. These effects of the state capacity regarding the sustainability of the state are expected to extend to the respective SWFs since states with a high state capacity have the possibilities of setting and controlling the mandate of their SWFs and, therefore, of using them as an instrument to achieve their objectives. In addition to the better regulatory environment and the available expertise, which facilitate the implementation of such a mandate, one can also assume that it should be easier for a SWF to find more and better trained administrative experts if the administration in a country is well prepared, at least if there is a certain degree of exchange and fluctuation between the state administration and employees of the fund that could be expected.

**Hypothesis 10.** *The higher the state capacity in the country of origin of a fund, the more sustainability criteria that it discloses into its mandate.*

### 2.9. IFSWF Membership (Santiago Principles)

SWFs have an interest in making investments in foreign countries [3,33] (p. 33). However, SWFs are often subject to scrutiny and skepticism in the countries where they want to invest [34]. In particular, democratic societies are often skeptical of accepting increasing foreign ownership of domestic companies, even more so if the foreign acquirers are state-controlled (so called "penalty of stateness") [35]. In order to prevent protectionist measures, SWFs have to make themselves legitimate. A possible way to do so is the establishment of good SWF governance practices, which ensure a sound management. In this vein, the International Working Group on Sovereign Wealth Funds (IWG) has devised the so-called Santiago Principles in 2008, a set of 24 principles with the aim to ensure good governance and public perception [36]. SWFs can commit to these principles and become a member of the International Forum of Sovereign Wealth Funds (IFSWF) on a voluntary basis to commit to operating as financial investors with a high level of transparency and disclosure. It is explicitly requested that any objectives other than economic ones, i.e., political objectives, are stated in the official investment policy and are disclosed to the public. Roughly speaking, the Santiago principles aim to align the norms of the SWFs with the countries receiving SWF investment [34]. An active promotion of sustainability is possible within the framework of the Santiago Principles if it is disclosed to the public as indicated by subprinciple 19.1 in the official document: "If investment decisions are subject to other than economic and financial considerations, these should be clearly set out in the investment policy and be publicly disclosed" [36] (p. 8). Since the Santiago Principles are intended to increase the legitimacy of the SWF and can, therefore, be expected to be positively associated with a high level of disclosure of the SWF leadership and governance, it is expected that funds, which have signed the Santiago Principles, are also likely to regard sustainability criteria in their investing to further increase their legitimacy. It is important to mention that there is no endogeneity problem using the Santiago Principles as one explanatory factor for the country results of the SDI. The operationalization of the governance score of the SDI is not dependent on whether a SWF has signed the Santiago principles, i.e., a reference to being a signatory of the Santiago Principles is not accepted as a basis for encoding the criteria of the SDI governance score. Rather, proof other than a reference to Santiago has to be available.

**Hypothesis 11.** *Funds that commit themselves to the Santiago Principles disclose more sustainability criteria into their mandate than funds that do not commit themselves.*

## 3. Materials and Methods

### 3.1. Operationalization of the Dependent Variable

In order to operationalize the disclosure of sustainability criteria by SWFs as the dependent variable of our research, we present in the following the construction of our SDI dataset (see for more details Supplementary A1–A4 and B2). We checked official documents of a selection of over 50 SWFs (the documents which were most recent as of March 2020) for the disclosure of a variety of, in total, 19 concrete sustainability criteria. We were able to cover the disclosure of sustainability criteria by SWFs at a current moment in time (year 2020). It was not possible to collect time series data since not all SWFs document which sustainability criteria were included into their mandate at what moment in time. Therefore, the analysis refers to the year 2020. We assigned the individual sustainability criteria to the categories "Environmental Aspects", "Social Aspects", "Governance Aspects", and "Economic Aspects" as shown in Table 1. Inspired by several more general sustainability measurements [8,37,38], the SDI covers a wide range of sustainability aspects linked to the investment behavior of SWFs. Regarding the environmental aspects, we look for commitments both to avoid environmentally harmful investments in fossil energies and air, water, and soil pollution as well as promises to make transformative steps toward more renewable energies and sustainable agriculture. Social aspects cover the promotion of human rights and poverty reduction as well as human capital/education building and promotion of sustainable health services not only in the country of origin of the fund, but also in developing countries worldwide. While governance criteria cover sound, responsible, and transparent management rules of the fund, they also include risk management obligations, checking whether financial as well as climate risks are considered appropriately. Finally, economic criteria look for commitments to invest in sustainable innovation and infrastructure projects and support for local economies and developing countries. Investments in unsustainable and unethical business practices should be excluded by the fund. While the Governance–Sustainability–Resilience (GSR) Scoreboard [5] (p. 17) also measures the disclosure of sustainability criteria by SWFs in one of its subcomponents (sustainability), the SDI shows stronger emphasis on the sustainability disclosure of SWFs, covering a broader, more differentiated set of sustainability criteria and goals. Besides some elements that also touch on aspects of sustainability in its subscores "Governance" and "Resilience", the GSR includes 10 elements in its subscore "Sustainability". In particular, element 16 (reference to SDGs) covers sustainability criteria evaluating whether a fund either claims to pursue SDGs in its annual report or is a "United Nations Principles for Responsible Investment" (UNPRI) signatory member [39] (p. 4). For the SDI, we decided not to use UNPRI membership as a criterion since there is a strong overlap of the six UNPRI principles [40] with some of our criteria, which we can grasp individually in a much more differentiated way. In addition, the UNPRI principles are not specifically tailored to SWFs. As a robustness check, we compared the results for funds, which are covered by our SDI dataset as well as by GSR, and we found strong and significant correlations between the SDI and the GSR Sustainability subscore ($r = 0.81$) and somewhat weaker correlations with the GSR total score ($r = 0.67$). SDI applies binary encoding in order to capture the presence or absence of the disclosure of the mentioned sustainability criteria. A "1" was coded if evidence is found that the criterion is fulfilled by the fund, i.e., if the fund obliged itself to fulfill the criterion or if concrete investment decisions related to the criterion could be found. A "0" was coded if neither of the previous cases were fulfilled or if there was an explicit statement that the fund does not regard the criterion. In every case in which a "1" was coded, a positive quotation justified the encoding. However, this was rarely possible for the categorization "0" since it is not common for SWFs to explicitly state that they do not fulfill a sustainability criterion. It has to be mentioned that this approach is prone to certain problems. A "0" means that no evidence for the fulfillment of a criterion was found but does not in itself mean that the criterion has never been regarded by the fund. For example, it might not have been treated with secrecy. On the other hand, a "1" does not provide information about the proportion of the investment decisions related to a

specific sustainability criterion and is rather intended to capture whether a basic regard for the criterion is present. Due to data-availability reasons, it is not possible to track the proportion of the investments related to the sustainability criteria with respect to the total investments. With our measurement, we cannot record the actual sustainability behavior of SWFs, but only their commitment to fulfill sustainability obligations. Accordingly, the encoded information captures whether evidence for a basic regard for the respective criteria is present or not. The encodings for the individual criteria as introduced previously are the basis for operationalizing the regard for sustainability on an aggregate level. Two trained research assistants carried out the coding independently and successively, based on the coding criteria shown in supplementary A3. Intercoder-reliability was sufficiently high (Cohen's Kappa well above the necessary benchmark) for all of the considered criteria, except for four criteria (Cohen's Kappa less than 0.40 for reduction in fossil energy, promotion of sustainable agriculture, promotion of sustainable infrastructure, and support local economy). The difference here can be explained since the present coder solely based the coding on official documents of the funds and followed somewhat stricter coding rules than the previous one. All results of the coding were cross-checked at the end by one of the authors. If a coding was still controversial, it was resolved through a discussion among the authors. An additive concept with equal weights is applied, i.e., for each SWF, the encodings for all sustainability goals are added to build the overall sustainability score and the score for the four sustainability pillars (environmental, social, governance, economic). Since no standard procedure is established [13], equal weighting prevents overemphasis on some sustainability criteria that would inherently conflict with a holistic concept of sustainability. In order to allow for a more in-depth analysis of different aspects of sustainability and to test the robustness of findings on a more basic level, the four sustainability pillars are also resolved.

**Table 1.** Considered criteria.

| Environmental Criteria | Social Sector Criteria | Governance Criteria | Economic Criteria |
|---|---|---|---|
| Promotion of renewable energy | Promotion of human rights | Board composition and leadership | Promotion of sustainable innovations |
| Reduction of fossil energy | Building human capital and/or education | Risk management (financial) | Promotion of sustainable infrastructure |
| Reduction of air, water, and soil pollution | Promotion of poverty reduction | Risk management (climate) | Prohibition to support unsustainable business practices |
| Promotion of sustainable agriculture | Promotion of sustainable health services | Business ethics | Support local economy |
| | | Transparency of investment | Investment to promote developing countries |
| | | Responsible ownership | |

### 3.2. Operationalization of the Independent Variables

As discussed above, we consider the structure of the fund, the size of the fund, the origin of the fund, the economic status of the country of origin, the natural resource wealth of the country of origin, the demography of the country of origin, the electoral democracy, the state capacity, and the IFSWF membership status as factors that might explain the disclosure of sustainability criteria by SWFs. For all these independent variables, data for the year 2016 are used. A time lag of the dependent variables of four years appears sensible to ensure that we can expect a causal effect on the dependent variable since most of the independent variables might likely have an effect with a certain delay. In addition, data restrictions on some of the independent variables mean that more up-to-date data were not accessible. We decided to collect all independent data for 2016 in order to create a uniform lag structure. The different independent variables, together with their reference name and the respective operationalization, are shown in Table 2 (for data access Supplementary B3–B7 and B11).

**Table 2.** Independent variables.

| Variable | Description of Operationalization | Variable Name in Regression | Data Obtained from |
|---|---|---|---|
| Structure of the fund (Pension Fund Mandate or other SWFs) | Binary encoding for the presence or absence of a pension fund mandate | no_pension_fund | Official SWF Documents |
| Size of the fund | Assets under management (AUM) in USD 1000 billions | AUM | [4,18] |
| Origin of Fund | Binary Encoding capturing whether the fund is commodity-based or non-commodity-based | origin_of_funding | [18] |
| Economic Status of Country of Origin | GDP/capita in USD 1000 | e_migdppc | [41,42] |
| Natural Resource Wealth of Country of Origin | Crude oil exports in 1000 ktoe | crude_oil_exports | [43] |
| Demography of Country of Origin | Percentage of population aged 0 to 14 years | pop_ages_0_14 | [44] |
| Electoral Democracy | Electoral democracy index on a scale of 0 to 1 | v2x_polyarchy | [41] |
| State Capacity | Governance effectiveness on a scale of −2.5 to +2.5 | e_wbgi_gee | [41,45] |
| IFSWF Membership (Santiago Principles) | Binary encoding for the presence or absence of the IFSWF membership | IFSWF_membership | [46] |

### 3.3. Multiple Linear Regression Analysis

In our empirical analysis, we calculate multiple linear regression models in order to account for the simultaneous presence of several different effects on the sustainability disclosure of SWFs (see supplementary B1). Regarding this issue, to our knowledge no quantitative analysis at this level of complexity has been submitted so far. Due to the expectations that we formulated for the influence factors being linear, multiple linear regression is appropriate to test our expectations. A further strong point for multiple linear regression analysis is that overcomplex models could lead to misleading implications due to overfitting on our sample. While the figures of merit for the models could be improved, it appears possible that such results cannot be generalized. We are aware that the number of included cases in relation to included variables is crucial for obtaining valid regression results. Taking into account the inherently limited number of SWFs and the complications in obtaining time series data, we can argue that we cover the maximum available number of cases. Although we do not achieve a full survey of all SWFs over time, we do cover a very large part of the SWFs that exist today. It has to be noted that our number of cases/number of variables ratio in our full models matches the basic requirements for regression analysis [47]. These are "at least 5 times more cases than IVs" [48] (pp. 128–129) and a "number of subjects N > 50 + m" [49], where m is the number of predictors. We also conduct regressions with reduced numbers of variables (fund-specific model; socioeconomic and political model) to validate the results (robustness check). Multiple linear regression analysis is implemented, using the statistical software *RStudio* (Version 1.2.5033) for the programming language *R* (Version 3.6.3). The variance inflation factors (VIF) are controlled in order to ensure that multicollinearity is not a major problem. Further, heteroscedasticity and the normal distribution of the residuals is tested. The formula applied for our full models is the following: sustainability_score ~ no_pension_fund + AUM + origin_of_fund + e_migdppc + crude_oil_export + pop_ages_0_14 + v2x_polyarchy + e_wbgi_gee + IFSWF_membership. For the fund-specific models, we only used the first three, and for the socioeconomic and political model, the last six variables.

## 4. Results

If we look at the empirical results regarding the disclosure of sustainability criteria into the mandate of SWFs, Figure 2 shows remarkable variance between the funds, mirroring the fact that across all funds, only 27 percent of the criteria were met. Looking at the scores for the four pillars regarding the environmental, social, governance, and economic dimension of sustainability, we also see significant differences (see Figure 3). It is noticeable that a relatively large number of criteria are met regarding the governance dimension. The proportion is 37 percent here, followed by 30 percent regarding the social sustainability dimension. In contrast to that, the proportion is significantly lower in the economic

(21 percent) and ecological (18 percent) sustainability dimension. The low coverage rate in total is based on a considerable number of funds (thirteen) for which we had to encode all criteria with a zero. These funds either do not regard any of the 19 considered sustainability criteria or do not disclose any information at all.

In addition, there appears to be another accumulation of funds, which regard between two and four criteria. Most of these funds disclose basic management and economic sustainability criteria (sometimes some social criteria) but lack commitments regarding the other dimensions of sustainable investment. Further, there is a third accumulation of funds with a broader coverage rate of around eight to ten criteria. While these funds at least partially cover the different pillars of sustainability, they show also gaps, mostly, but not exclusively, in the ecological sustainability dimension.

Only five funds disclose more than twelve sustainability criteria, while no fund in our sample was able to fulfill all possible 19 criteria at once. A more in-depth analysis demonstrates that even the best performers—the New Zealand Superannuation Fund and the Ireland Strategic Investment Fund (16 disclosed criteria) and the Government Pension Fund of Norway (15 criteria disclosed)—show some missing with regard to the economic and environmental (and for two of them, also the social) dimension. In total, only a small minority of 14 funds (presented in Table 3) disclose more than half of the criteria while, at the same time, covering all four sustainability pillars, which could qualify them as followers of an overarching sustainability strategy.

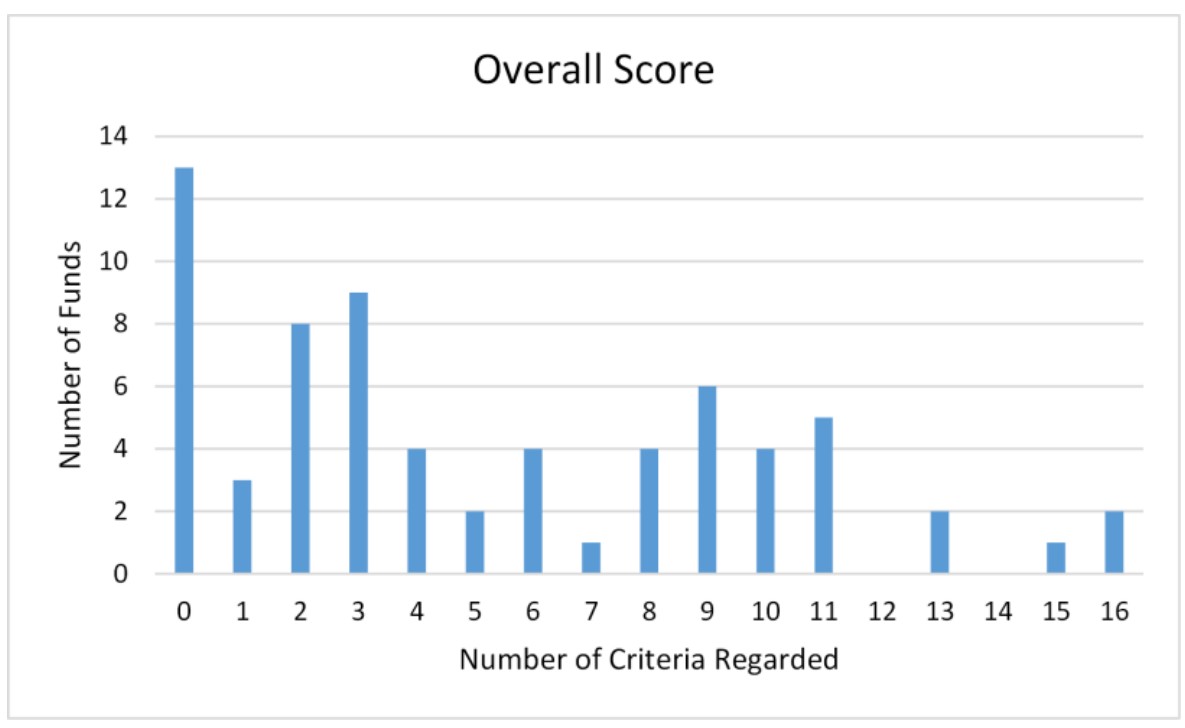

**Figure 2.** Distribution of the overall sustainability score.

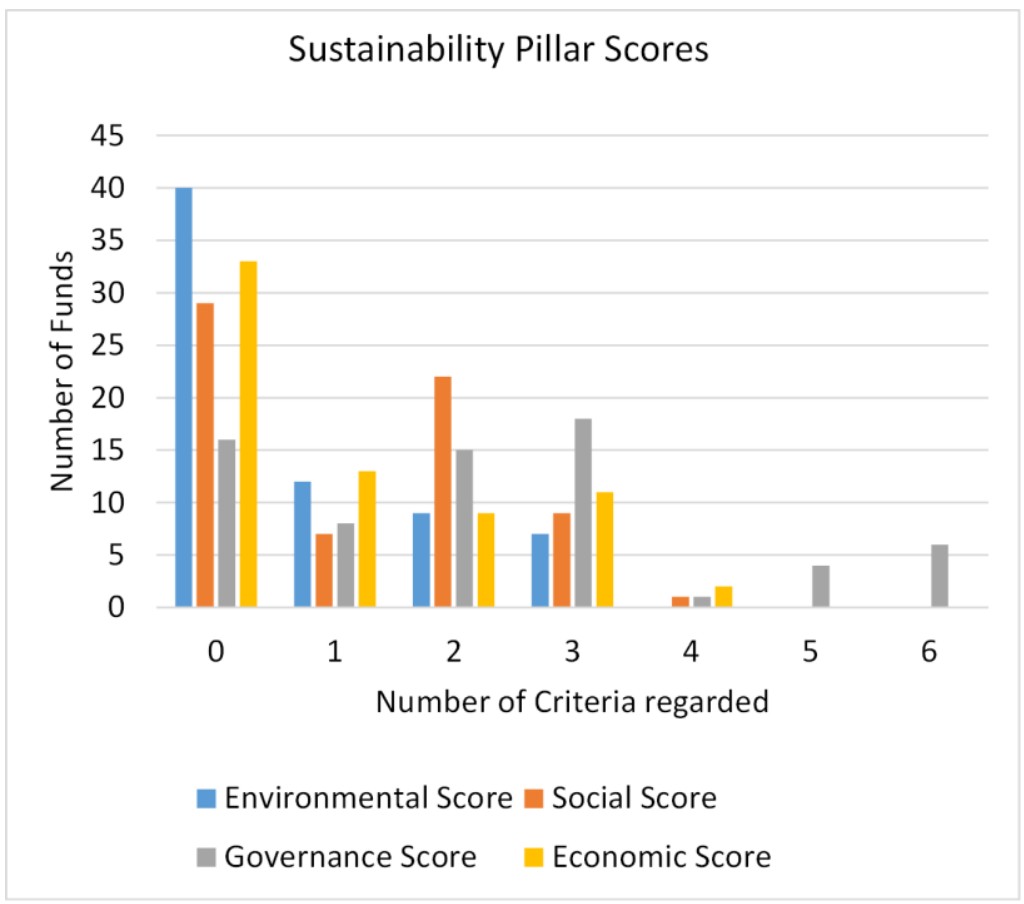

**Figure 3.** Distribution of the score for the four sustainability pillars (environmental, social, governance, economic).

**Table 3.** Best performing SWFs.

| SWF | Environmental Score | Social Score | Governance Score | Economic Score | Overall Score |
|---|---|---|---|---|---|
| New Zealand Super Fund | 3 | 3 | 6 | 4 | 16 |
| Ireland Strategic Investment Fund | 3 | 4 | 6 | 3 | 16 |
| Government Pension Fund Global of Norway | 3 | 2 | 6 | 4 | 15 |
| Fourth Swedish National Pension Fund | 3 | 2 | 6 | 2 | 13 |
| Temasek Holdings | 3 | 2 | 6 | 2 | 13 |
| Abu Dhabi Investment Authority | 1 | 2 | 5 | 3 | 11 |
| Australia Future Fund | 1 | 3 | 4 | 3 | 11 |
| Khazanah Nasional Berhad | 1 | 3 | 5 | 2 | 11 |
| Public Investment Fund of Saudi Arabia | 3 | 2 | 3 | 3 | 11 |
| Samruk-Kazyna JSC (S-K JSC) | 2 | 3 | 3 | 3 | 11 |
| Mubadala Investment Company | 2 | 2 | 3 | 3 | 10 |
| Nigeria Sovereign Investment Authority | 2 | 3 | 3 | 2 | 10 |
| Senegal Strategic Investment Fund | 3 | 3 | 2 | 2 | 10 |
| Alberta Heritage Savings Trust Fund | 2 | 2 | 3 | 3 | 10 |

If we look at the regression results for the overall sustainability score and the environmental, social, governance, and economic score (Table 4), we see that our full model specifications that include all independent variables have relatively good explanatory power. This applies particularly to the full regression model for the governance score (adjusted $R^2$: 0.650), while a significantly lower part of the variance can be explained for the environmental score (adjusted $R^2$: 0.213). The other models are lying in between (overall score, adjusted $R^2$: 0.484). The adjusted $R^2$ of the model specification with reduced samples of independent variables (fund-specific model; socioeconomic and political model) are expected to be lower. Although one could argue that a crucial part of the explanatory power of the overall score model is driven by the results of the governance score model, a detailed inspection shows that this does not hold with regard to the role of all influencing factors. While high assets under management (AUM) are significantly positive correlated with a high governance score, this is not the case for the overall score. With regard to the different explanatory factors, we see significant positive correlations with high scores across all the full models for state capacity (e_wbgi_gee) and IFSWF membership (IFSWF_membership). This also holds true for the models with reduced number of variables (model 2, 5, 8, 11, 14). The percentage of people younger than 14 years (pop_ages_0_14) are significantly positively associated in the full models with the social, governance (not in the reduced model 8), and overall score, not meeting the significance criteria for the environmental and economic score. Contrary to this, the high natural resource wealth of the country of origin (crude_oil_export) is, however, negatively associated with all scores, only becoming significant for the environmental and economic score (not in the reduced models 2 and 11). Commodity-based funds (origin_of_funding) are, in all full models, significantly positively associated with all, except for the governance score (in reduced model it is), while, once again, contrary to this, funds with high assets under management (AUM) are in a significant, positive relationship with a high governance score and a negative relationship to all the others, which is not significant here. Besides the significant correlations, we also find remarkable non-correlations. Notably, the proxy for electoral democracy (v2x_polyarchy) does not have an explanatory value in any of the models. In addition, the proxies for the characteristic as a (non-)pension fund (no_pension_fund) and the economic status of the country of origin (e_migdppc) do not have any explanatory power.

As additional information and a robustness check, we added in supplementary B9 for the socioeconomic_and_political models and in supplementary B10, for the full model regressions, including the indicator "percentage of people older than 65 years" instead of "percentage of population aged 0 to 14 years". This alternative indicator does not, however, become significant in any model specification.

If we take a closer look at the specific situation of those 14 funds that, based on their disclosure rate, qualify as followers of an overarching sustainability strategy, we see that most of them fit very well into the correlation patterns identified in our regression analysis for all SWFs. The majority of them are rather large, non-commodity-based funds situated in rich countries that can rely on a high state capacity (exceptions are only the SWFs from Nigeria and Senegal). While a broad majority are IFSWF members, many, but by no means all of them, are located in democratic countries. In contrast to the relationships shown in the regression analysis for all SWFs, most of them are situated in countries with a rather old population.

**Table 4.** Regression analysis result for the overall sustainability score and the environmental, social, governance, and economic score.

| | Dependent Variable: | | | | | | | | | | | | | | |
|---|---|---|---|---|---|---|---|---|---|---|---|---|---|---|---|
| | environmental_score | | | social_score | | | governance_score | | | economic_score | | | overall_score | | |
| | Fund-specific Model (1) | Socioeconomic and Political Model (2) | Full model (3) | Fund-specific Model (4) | Socioeconomic and Political Model (5) | Full Model (6) | Fund-specific Model (7) | Socioeconomic and Political Model (8) | Full Model (9) | Fund-Specific Model (10) | Socioeconomic and Political Model (11) | Full Model (12) | Fund-specific Model (13) | Socioeconomic and political Model (14) | Full Model (15) |
| no_pension_fund | 0.151 (0.483) | | −0.178 (0.461) | 0.298 (0.536) | | −0.204 (0.484) | −0.284 (0.769) | | −0.822 (0.531) | 0.135 (0.590) | | −0.192 (0.551) | 0.299 (2.026) | | −1.397 (1.620) |
| AUM | −0.193 (0.626) | | −0.595 (0.656) | −0.006 (0.695) | | −0.449 (0.689) | 3.405 ** (0.997) | | 1.939 * (0.756) | 0.168 (0.765) | | −0.497 (0.784) | 3.373 (2.627) | | 0.398 (2.306) |
| origin_of_funding | 0.686 * (0.279) | | 0.776 * (0.315) | 0.887 ** (0.310) | | 0.896 ** (0.331) | 0.918 * (0.444) | | 0.626 (0.363) | 0.849 * (0.341) | | 0.892 * (0.377) | 3.340 ** (1.170) | | 3.190 ** (1.108) |
| e_migdppc | | −0.007 (0.007) | 0.004 (0.008) | | 0.001 (0.008) | 0.012 (0.008) | | −0.001 (0.010) | 0.009 (0.009) | | −0.007 (0.009) | 0.005 (0.010) | | −0.014 (0.028) | 0.030 (0.028) |
| crude_oil_exports | | −0.003 (0.002) | −0.004 * (0.002) | | −0.001 (0.002) | −0.002 (0.002) | | −0.0001 (0.002) | −0.001 (0.002) | | −0.003 (0.002) | −0.004 * (0.002) | | −0.007 (0.006) | −0.012 (0.006) |
| pop_ages_0_14 | | 0.042 (0.026) | 0.051 (0.025) | | 0.072 * (0.028) | 0.082 ** (0.027) | | 0.057 (0.035) | 0.078 * (0.029) | | 0.045 (0.030) | 0.055 (0.030) | | 0.216* (0.097) | 0.267 ** (0.089) |
| v2x_polyarchy | | −0.049 (0.493) | −0.094 (0.531) | | −0.284 (0.532) | −0.317 (0.558) | | −1.103 (0.661) | −0.698 (0.612) | | −0.179 (0.571) | −0.089 (0.634) | | −1.615 (1.846) | −1.198 (1.865) |
| e_wbgi_gee | | 0.762 ** (0.268) | 0.622 * (0.268) | | 0.855 ** (0.290) | 0.730 * (0.282) | | 1.633 *** (0.360) | 1.426 *** (0.309) | | 0.947 ** (0.311) | 0.823 * (0.320) | | 4.197 *** (1.005) | 3.601 *** (0.942) |
| IFSWF_membership | | 0.720 ** (0.269) | 0.650 * (0.276) | | 0.840 ** (0.291) | 0.714 * (0.290) | | 1.982 *** (0.361) | 1.749 *** (0.318) | | 0.860 ** (0.312) | 0.679 * (0.330) | | 4.402 *** (1.008) | 3.793 *** (0.970) |
| Constant | 0.402 (0.486) | −0.859 (0.697) | −1.406 (0.789) | 0.668 (0.539) | −1.176 (0.753) | −1.790 * (0.829) | 1.812 * (0.774) | −0.174 (0.936) | −0.702 (0.909) | 0.679 (0.594) | −0.718 (0.808) | −1.375 (0.943) | 3.561 (2.038) | −2.927 (2.614) | −5.274 (2.772) |
| Observations | 61 | 65 | 58 | 61 | 65 | 58 | 61 | 65 | 58 | 61 | 65 | 58 | 61 | 65 | 58 |
| R$^2$ | 0.096 | 0.219 | 0.337 | 0.127 | 0.273 | 0.412 | 0.235 | 0.539 | 0.705 | 0.101 | 0.243 | 0.351 | 0.156 | 0.417 | 0.565 |
| Adjusted R$^2$ | 0.049 | 0.138 | 0.213 | 0.081 | 0.198 | 0.302 | 0.194 | 0.491 | 0.650 | 0.054 | 0.165 | 0.229 | 0.112 | 0.357 | 0.484 |
| Residual Std. Error | 1.018 (df = 57) | 0.975 (df = 58) | 0.934 (df = 48) | 1.130 (df = 57) | 1.053 (df = 58) | 0.981 (df = 48) | 1.622 (df = 57) | 1.309 (df = 58) | 1.076 (df = 48) | 1.244 (df = 57) | 1.129 (df = 58) | 1.116 (df = 48) | 4.272 (df = 57) | 3.653 (df = 58) | 3.281 (df = 48) |
| F Statistic | 2.022 (df = 3; 57) | 2.713 * (df = 6; 58) | 2.710 * (df = 9; 48) | 2.760 (df = 3; 57) | 3.626 ** (df = 6; 58) | 3.743 ** (df = 9; 48) | 5.822 ** (df = 3; 57) | 11.305 *** (df = 6; 58) | 12.771 *** (df = 9; 48) | 2.136 (df = 3; 57) | 3.105 * (df = 6; 58) | 2.885 ** (df = 9; 48) | 3.511 * (df = 3; 57) | 6.916 *** (df = 6; 58) | 6.931 *** (df = 9; 48) |

Note: Signif. codes: '***' $p < 0.001$, '**' $p < 0.01$, '*' $p < 0.05$.

## 5. Discussion

The results of our empirical investigation confirm the assumptions formulated in Hypotheses H1 and H2. To date, only very few SWFs have implemented an overarching sustainability strategy, disclosing a broad majority of sustainability criteria into their mandate. The distribution of the overall score indicates that sustainability is not yet a top priority of most SWFs. This, however, does not mean that disclosure of certain sustainability criteria is totally absent looking at SWFs worldwide. We see patterns of selective disclosure, focusing on governance criteria helping the fund to make themselves more credible and legitimate for regulators and investors, generating direct benefits for investment opportunities and the fund's ability to act (present-day oriented economic optimization strategy).

However, there is still a considerable variance between the funds regarding their willingness to disclose sustainability criteria into their mandate. As our analysis shows, this variance depends only partially on fund-specific characteristics. Interestingly, the pension fund status appears to have no significant effect on disclosure in any of our models. This finding contradicts the initial expectation (H3) that pension funds are less sustainable due to the pressure to generate revenue in shorter timeframes for their beneficiaries. An explanation for this finding could be that the sample only contains a small number of pension funds. Therefore, there might be not enough variance in this variable for explaining the differences in the outcome. With regard to the AUM of an SWF, we find only a significant positive association with the governance score, while for the other scores, the relation is insignificant (and negative). Although larger funds seem to find it easier to adopt management rules, being subject to stronger scrutiny from the political system and the public due to their size does not seem to affect their willingness to take further the sustainability aspect more seriously (contradicting H4). Only the origin of funding of an SWF seems to have an important impact on its willingness to commit itself to sustainability investment criteria. As theoretically expected (H5), commodity-based funds seem to have a harder job of distancing themselves from unsustainable business practices regarding their investment requirements. This applies to all sustainability dimensions measured here with the exception of the management score.

Socio-economic factors of the country of origin also play a subordinate role for the formal sustainability behavior of a SWF, with one important exception. Therefore, the economic development level of the home country, measured by the national GDP per capita, does not seem to have any significant effect (contradicting H6). Besides the debate of whether the GDP per capita is a good metric to measure national wealth [50], our regression results go in line with the argument that SWF seems to be able to decouple themselves more than other economic agents from the economic situation of their home country. Thus, a high economic development level does not guarantee that a SWF follows a strict sustainability strategy. The assumption that SWFs from resource-rich oil countries are less willing to commit themselves to sustainability standards (H7) even tends to be refuted. At least for the environmental and economic sustainability dimensions, we find significant relations that point to the opposite direction. Whether one can interpret this as proof that in oil exporting countries, the need to become independent from oil export dependence due to oil being a depletable natural resource puts specific pressure on SWFs to invest in a more sustainable way should be part of further research effort. The fact that we find strong, significant, positive relations between the demographic characteristics of a fund's home country and its formal sustainability efforts provides arguments that SWFs (have to) consider long term expectations and interests of key segments of the population. In line with our theoretical expectations (H8), we see that funds located in a country with a high share of young people are more willing to include (especially social) sustainability criteria into their mandate. This result is remarkable, although for the funds with the highest disclosure rate, it is not very decisive. Although young citizens have no direct lobby influence on political decision-making (as they are, for instance, not yet eligible to vote) decision-makers of SWFs in young societies seem to be more willing to orientate their

fund to the long-term interests of the younger generation. In addition, they seem to be able to shield themselves from the direct political influence of older segments of the population, which, due to their shorter remaining lifespans, may want them to generate, above all, short-term revenues instead of sacrificing revenue for more sustainable investing.

The ability of SWFs to act independently from direct public pressure may also be the reason why the degree of democratization in a home country is not significantly related to the actions (or non-actions) of a fund when proposing sustainability obligations. This observation, contradicting H9, holds true not only for the overall sustainability score, but also for the partial scores for all four pillars. Even so, four of the top five SWFs regarding our sustainability measurement originate from democratic countries; the top ten contain a substantial share of funds from autocratic countries. Further, there are a number of SWFs from democracies that hardly regard any sustainability criteria. Therefore, in our regression analysis, we do not find a clear positive democracy effect. Rather, the relationship in all of our regression models is negative; even so, it does not reach a sufficient significance level. These results are contradicting with our theoretical expectations and the observations of Megginson et al. [5]. By looking at the rating results of the funds in their sample, they see that most of the funds with a high GSR score are located in democracies, while only a few funds in democracies have very low score ratings. Concluding from this, they argue that "the more democratic the sponsoring society, the better governed, more sustainable, and more resilient their state-owned investment funds tend to be" [5] (p. 18). However, these descriptive findings are not underpinned by a quantitative analysis. If we evaluate the results for funds, which are covered by our SDI dataset as well as by GSR (see for data supplementary B8), we find no strong correlation between GSR Sustainability subscore and the electoral democracy index. The same is observed for our SDI score. The correlation between the GSR total score and the electoral democracy index are somewhat stronger. All of this indicates that the patterns regarding regime type and sustainability disclosure of SWFs might be similar, using the GSR or our SDI measurement. The insignificant results may also reflect opposing effects caused by democratic processes based on free elections. Besides its innovation potential for sustainable transformation processes, the inherent democratic bias caused by short-term voting cycles is strong [11,23]. Therefore, it remains an open question as to whether democracy enhances the ability of a society to think in extended timescales and provide institutional capacities to stick to long-term strategies and commitments.

Contrary to the weak linkage regarding the regime type, our regression results show that there is, as theoretically expected (H10), a strong positive link between the high state capacity of a country of origin and the disclosure of sustainability criteria by a SWF. This consistent finding regarding all pillars of sustainability measured here stresses the importance of functional state capacity as a key precondition for the long-term orientation of a state, eventually rubbing off on SWFs, even though they are not under direct state control. One can argue that an advanced and well-organized state might have a higher regard for sustainability due to its focus on long-term self-preservation, which a dysfunctional state is unable to follow or implement. A functioning regulatory environment seems to make it easier for an SWF to set long-term goals and commitments, even though review by functional state institutions could also be a double-edged sword for sustainability commitments by a fund since it puts pressure on it to take its commitments seriously. On the one hand, it might help to prohibit misleading reports, selective disclosing, and "greenwashing" by the fund [48]. On the other hand, it can also reduce the motivation of a SWF to set far-reaching sustainability goals and requirements.

Finally, similar to the regression results for the domestic factor state capacity, IFSWF membership as an international commitment of a SWF has a significant positive association in all regression models with the disclosure of sustainability criteria. Although theoretically expected (H11), this consistent result for all pillars of sustainability is remarkable since the Santiago Principles are mainly intended to ensure sound management and governance practices of the funds. The goal is to make SWFs legitimate and to avoid

investment-recipient countries from taking protectionist measures against the SWFs [33]. The significant positive association of the IFSWF membership on the governance pillar is somehow a logical consequence even though the criteria of our sustainability measurement differ from those enumerated in the IFSWF. However, there is no explicit demand to regard societal or environmental aspects in the IFSWF statement. Funds participating in the IFSWF stand under international public pressure to make their investment behavior more transparent. In doing so, IFSWF membership could have a positive effect on the publication of further obligations that are not explicitly mentioned in the Santiago Principles. Another possible explanation for this positive association between IFSWF membership and a high level of disclosure might be the underlying factor that both might help a fund to make itself more legitimate against other investors and state regulators. Therefore, we may see here no causal effect but more of a coincidence between the two factors.

## 6. Conclusions

The bottom line of this study and its first important research contribution is that the vast majority of SWFs worldwide to date have failed to disclose sustainability criteria into their mandate to the extent that it can be understood as part of an overarching sustainability strategy. This indicates that SWFs, so far, are not the vehicles for a sustainable realignment of investment regimes that they could be. With the exception of very few top-runner funds, disclosure takes place selectively and is focused on individual criteria (especially those regarding sustainable governance).

However, our results also show that sustainability aspects are, if not an integral, at least a partial aspect of many SWF business models in our days. Although one could argue that this selective disclosure takes place not out of conviction but for opportunistic reasons, i.e., creating a sense of legitimacy for the respective SWF, many funds commit themselves at least to a certain degree. Besides permanent incentives for "greenwashing", this creates room for real improvements since stronger orientation on sustainable investment and (ecological) risk management might help to avoid financial risks caused by unsustainable investing behavior (financial self-interest of the fund) and also be part of a long-term survival strategy.

A second important finding of our analysis is that different factors are significantly related to an SWF's commitment to sustainability. It is interesting that many of these factors can be politically influenced, at least in the medium term. For the disclosure of sustainability criteria by a fund, the socio-economic conditions of its country of origin, which can hardly be changed in the short term, do not seem to play a central role (with the exception of demography). The same is true for the size and nature of a fund (with the exception of its original source of funding). Rather, the regulatory environment, a state capable of acting, and the international involvement of a fund seem to play an important role. A high degree of democratization in its home country, however, does not seem to be a mandatory prerequisite for the strong sustainability commitment of a SWF. It should be emphasized that from the perspective of developing countries with a rather young population, none of the structural factors stand in the way for a strong sustainability commitment by a SWF.

The aim of our study was to shed light on the sustainability disclosure in the mandate of SWFs and to conduct a systematic analysis regarding explanatory factors for their behavior. As a first quantitative study on this issue, it comes with certain limitations. This relates firstly to the time period under investigation. Since longer-term time series data are not available for the dependent variable, we focus on the current situation in the year 2020. Even though we included a large part of the SWFs worldwide in our analysis, we could not include the whole universe of possible cases. In particular, very small SWFs we could not consider due to missing data.

Since we are focusing on published commitments by the funds, it is not possible to say something about the concrete sustainability investments made by SWFs, their proportion or growth. Even though research on concrete investment decisions by SWFs is

very important, it is still at its infancy and, to date, very selective since precise evaluation of the related numbers and detangling of the different sustainability aspects is a very complex task [7]. With our approach, we are able to record sustainability dimensions in a differentiated way and to trace the obligations that the funds have formally given. What we cannot see, however, is what concrete investment decisions are made by a SWF to fulfill these obligations. In doing so, our approach might be prone to selective disclosure and "greenwashing" attempts by SWFs in the case that the mandates are only paper tigers [51]. Although our study already documents significant deficits at the level of formal obligations, it could even overestimate the actual sustainability orientation of SWFs. "Greenwashing" seems particularly likely if compliance with formal commitments is difficult to control and goes hand in hand with high costs.

Our investigation, which has already identified significant gaps in formal commitments by SWFs, should be supplemented by investigations that look at concrete investment decisions of SWFs. SWF are in many cases based on the sale of natural resources. When investigating the potential of SWFs as sustainability instruments, it might also be important to control for the share of revenue from natural resources in a country, which is redirected to SWFs. For achieving weak sustainability, it is enough that resource revenues from the sale of natural resources are used for sustainable investments. It is therefore important that a substantial share of these revenues is directed to sustainable SWFs in order to obtain a meaningful contribution. Even SWFs which invest in a sustainable way might be ineffective if they are small in relation to the resource sales of their country of origin.

With regard to the explanation of the varying disclosure rates, we find significant correlations in our models, which appear to be plausible both in terms of their direction and the magnitude of their coefficients. However, the overall explanatory power of our models for some of the sustainability dimensions is limited. The adjusted R-squared is highest for the governance score model. This shows that the explanatory framework describes the governance score result best and applies to the highest share of SWFs. The environmental, the social, and the economic score models yield comparatively lower values for the adjusted R-squared. This reveals that the explanatory framework is subjected to limitations in explaining these aspects and that disturbances are superimposed over the explanatory framework. Selective disclosure of sustainability criteria and random influences might be strong obstacles to finding appropriate models that do not overfit. However, in further research it is important to see whether it is possible to find other and maybe better factors that help to explain the variance here. These should include, on the one hand, more fund-specific factors based on in-depth (qualitative) investigations regarding their internal organization and structure as well as corporate philosophy and mindset. On the other hand, the external embedding of SWFs into networks, formal and informal arrangements as well as actor constellations should be examined.

In addition, it is necessary to seek a better understanding of the causal links and mechanisms explaining the relationships and non-relationships between certain factors and disclosure rates. What are the concrete mechanisms that link a high state capacity to a high discloser rate? What are the reasons for the missing link between the political regime type and sustainability orientation of SWFs? Are the short-term oriented institutional settings in democracies (permanent election campaigns) or weak demand for sustainability in democratic societies responsible for the missing democracy effect? To gain more in-depth insights regarding these relationships and the underlying mechanisms, qualitative case studies of individual funds will be helpful. Based on this, we might better see what incentives are necessary for SWFs to strengthen their commitment for sustainable investment.

Our quantitative analysis already provides some first indications. Mechanisms that help SWFs to professionalize and to strengthen insight that unsustainable investments are a financial risk and could even undermine the foundations which SWFs are based on in the long run should be strengthened. In addition, domestic and international pressure should be increased so that sustainability is seen as necessarily related to the legitimacy of a SWF. If only those SWFs with an overarching sustainability strategy covering all of the individual

sustainability criteria were perceived as legitimate, there would be a strong incentive to introduce such a strategy. It can be expected that SWFs adjust their strategy accordingly. Basically, the legitimacy of a SWF becomes relevant at two pressure points, namely, when reporting to the mandate setter of the SWF and when looking for investment opportunities. The mandate setter, i.e., the government of the country of origin, can set the mandate so as to include sustainability criteria, and investment recipient countries could possibly take protectionist measures against SWFs without sustainability strategies. However, both of these pressure points appear to be difficult to utilize for the implementation of a sustainability strategy since they are highly individualized among the countries of origin of SWFs and investment recipients. It appears that the best way is to have similar standards for, ideally, all SWFs. It is likely that SWFs compare their performance against other SWFs of comparable size and that they aim to avoid comparatively lower revenues. Therefore, it appears viable to use the existing IFSWF framework and to extend it to promote SWF sustainability strategies. Making an amendment to the IFSWF statement, requiring a credible sustainability strategy to be disclosed and abided by is one possible method to achieve this. The advantage would be that a substantial share of SWFs are already IFSWF members and the IFSWF provides a forum for SWFs to directly discuss their common norms and practices. If a substantial share of the IFSWF members requested introducing a mandatory overarching sustainability framework, there would be a strong pressure for the other funds to follow this development, i.e., since leaving the IFSWF would be a strong disadvantage. This could possibly be initiated by the countries of origin of the top-runner SWFs. A possible problem remains in that the IFSWF might be misused for selective reporting. Therefore, a mechanism for penalizing not fulfilling the sustainability strategy and for increasing the share of investments intended to promote sustainability should be installed.

**Supplementary Materials:** Supplementary A1–A4 and B1–B11 are available online at https://www.mdpi.com/article/10.3390/su13105565/s1.

**Author Contributions:** Conceptualization, S.J.S. and S.W.; methodology, S.J.S. and S.W.; software, S.J.S.; validation, S.J.S.; formal analysis, S.J.S.; investigation, S.J.S. and S.W.; resources, S.J.S. and S.W.; data curation, S.J.S.; writing—original draft preparation, S.J.S. and S.W.; writing—review and editing, S.J.S. and S.W.; visualization, S.J.S. and S.W.; supervision, S.W.; project administration, S.W. Both authors have read and agreed to the published version of the manuscript.

**Funding:** This research received no external funding.

**Institutional Review Board Statement:** Not applicable.

**Informed Consent Statement:** Not applicable.

**Data Availability Statement:** The data presented in this study are openly available at https://www.mdpi.com/article/10.3390/su13105565/s1.

**Acknowledgments:** We want to thank Luma Rahman and Sophia Schmid for supporting data preparation.

**Conflicts of Interest:** The authors declare no conflict of interest.

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
