# Peer review of "Sovereign Wealth Funds as Sustainability Instruments? Disclosure of Sustainability Criteria in Worldwide Comparison"

_sustainability, doi:10.3390/su13105565_

Round 1
Reviewer 1 Report
First of all, the paper “Sovereign Wealth Funds as Sustainability Instruments? Disclosure of Sustainability Criteria in Worldwide Comparison” aims and scope match those of Sustainability, so the paper is adequate for this journal. However based on my opinion it needs substantial improvements to be considered for publication in Sustainability. I would suggest a series of changes that in my opinion would improve the paper, in special for the reader.
- Rewrite Abstract - Abstract should cover five main elements, introduction, problem statement, methodology, contributions and results.
- I suggest the authors to improve the introduction section. Authors should better highlight the objective of their work and to what extent it contributes to close a gap in the existing literature and/or practice. What is the innovative value of the contribution proposed by the authors?
- Add flowchart of proposed methodology and organize the rest of the paper in accordance to the stages defined in the flowchart.
- The results are good but validation section is missing. How we can judge about these results? Comparisons with existing methods from the literature is missing.
- The conclusion section. The authors need to clearly provide several solid future research directions. Clearly state your unique research contributions in the conclusion section. Add limitations of the model. No bullets should be used in your conclusion section.
Reviewer 2 Report
The article takes up a new issue. The authors see a justifiable need to study the inclusion of sustainable development by not researching institutions such as the SWFs. It is right to take up this issue because states and their societies expect businesses to respect UN SDGs, and the question is whether units subordinate to states meet these criteria. Therefore, I believe that undertaking such an analysis was by all means desirable. The activities of funds are often surrounded by an atmosphere of secrecy and conjecture about the state nature of their property, so any form of research helps to better understand the role of these funds in the global economy and meet the transparency criteria which also belongs to the sustainability idea.
I have a few minor notes for the article:
a. What the authors understand by “our days” (verse 60). I believe that this should be clarified for a few issues. First, Santiago principles were developed in 2008, and UN SDGs in 2015. Changes regarding sustainable development are happening very quickly, so please explain this term – what time you are interested in. Secondly, the article will be read in a few years and it may give rise to the belief that it is about this future time.
b. The size of the studied population and the selection criteria for the sample also require a more detailed explanation. Information that there was selected more than 50 funds, is rather general and, in my opinion, requires clarification (specifically how many funds were analyzed), especially that the number of funds is countable. Do I correctly understand that 14 funds were examined because they met at least one of the 19 criteria? Please clarify.
Reviewer 3 Report
Summary
The paper presents the Sustainability Disclosure Index (SDI), an original index to investigate the disclosure of 19 ESG criteria for a selection of over 50 SWFs. The main conclusion seems to be that, since SWFs can be expected to be influenced by the respective countries political and economic systems and long-term orientation is a strong precondition for its preservation, identifying favorable conditions for a higher commitment of SWFs could help to initiate pathways to become functional sustainability instruments.
Comments
- The paper deals with an important topic. The research questions seem to be appropriate and somewhat original in the ESG discussion. However, there are some problems from a methodologic point of view.
- The way the dependent variable is build seems to be unusual since authors do not collect different data for different years, but (it seems to me) only one observation for several years. On the other hand, the independent variable seems to be collected only for 2016. In other word it’s not so clear which years the analysis is referring to; it’s not so clear too the reason why Authors seems to use 2016 data for independent variables. I would expect Authors to better discuss the point and to support their methodologic strategy with the literature contribution.
- Moreover, within Authors research strategy, there are some problems with the significance of the dependent variable that may influence the results and their interpretation and, “due to data availability reasons”, it is not possible to track the proportion of the ESG investments with respect to the total investments”. I would expect Authors to better discuss the point and to clear the consequence on paper results.
- A (potential) main problem with the paper research strategy seems to be that the regression is based on too few data. We have, in fact, only 58 observations. It’s important Authors to clear the relationship between the sample used and the universe of SWFs in order to assess the statistical relevance of their analysis.
- Generally speaking, with a very small number of observations it’s not wise to have a great number of hypothesis to test and (as consequence) a great number of independent variables to work with. I think Authors should select a lower number of variables servicing more focused research questions.
- I believe it would be necessary (if it’s impossible to broaden the time spectrum and the observations number) to change the research strategy by running a more focused econometric analysis and a broader qualitative discussion of results.
Round 2
Reviewer 1 Report
The authors have addressed the point of my concern. I am happy with their corrections. Hence, I would like to recommend this manuscript to be published.
Reviewer 3 Report
the paper has been improved and can be published.